# Reversed Mg-Based Smectites: A New Approach for CO_2_ Adsorption

**DOI:** 10.3390/nano14181532

**Published:** 2024-09-21

**Authors:** Francisco Franco, Juan Antonio Cecilia, Laura Pardo, Salima Essih, Manuel Pozo, Lucía dos Santos-Gómez, Rosario M. P. Colodrero

**Affiliations:** 1Departamento de Química Inorgánica, Cristalografía y Mineralogía, Facultad de Ciencias, Campus de Teatinos s/n, Universidad de Málaga, 29071 Málaga, Spain; ffranco@uma.es (F.F.); lpardo@uma.es (L.P.); salima@uma.es (S.E.); ldsg@uma.es (L.d.S.-G.); 2Departamento de Geología y Geoquímica, Facultad de Ciencias, Campus de Cantoblanco, Universidad Autónoma de Madrid, 28049 Madrid, Spain; manuel.pozo@uam.es

**Keywords:** acid treatment, microwave, smectites, PCHs, CO_2_ adsorption

## Abstract

Addressing climate change requires transitioning to cleaner energy sources and adopting advanced CO_2_ capture techniques. Clay minerals are effective in CO_2_ adsorption due to their regenerative properties. Recent advancements in nanotechnology further improve their efficiency and potential for use in carbon capture and storage. This study examines the CO_2_ adsorption properties of montmorillonite and saponite, which are subjected to a novel microwave-assisted acid treatment to enhance their adsorption capacity. While montmorillonite shows minimal changes, saponite undergoes significant alterations. Furthermore, the addition of silica pillars to smectites results in a new nanomaterial with a higher surface area (653 m^2^ g^−1^), denoted as reversed smectite, with enhanced CO_2_ adsorption capabilities, potentially useful for electrochemical devices for converting captured CO_2_ into value-added products.

## 1. Introduction

Over the past century, global energy demand has risen sharply due to population growth and economic development. This increased demand spans across coal, oil, natural gas, nuclear power, and renewable energy sources. However, the heavy reliance on fossil fuels remains the primary driver of global climate change and environmental challenges, largely due to the emission of greenhouse gases [1]. Carbon dioxide (CO_2_) is responsible for over 60% of total emissions causing global warming. Atmospheric CO_2_ concentrations have risen from 300 ppm before the Industrial Revolution to over 400 ppm today [2]. This increase has contributed to a global average temperature rise of approximately 6 to 8% over the past century [3]. To address these challenges, the global energy system needs to shift to cleaner sources and explore methods like adsorption on porous media for capturing and storing CO_2_ emissions. The capture process aims to concentrate the CO_2_ stream for future geological storage or chemical use rather than merely storing the CO_2_ with an absorbent [4,5,6].

Clay minerals possess high porosity primarily due to their small particle size and unique crystallochemical properties, which make them highly effective for CO_2_ adsorption [7]. Within the clay mineral family, smectite and sepiolite groups are particularly noted for their superior adsorption capabilities. These minerals belong to the subgroup of phyllosilicates, whose physical and chemical characteristics are influenced by their fine grain size (<2 μm) [8]. These phyllosilicates exhibit highly intriguing physicochemical properties, such as high cation exchange capacity and high specific surface area, which, together with their low economic cost, make them very interesting materials for addressing diverse environmental challenges [9]. Smectites include both dioctahedral and trioctahedral varieties, determined by the occupancy of octahedral positions. One of the most abundant dioctahedral smectites is the montmorillonite, (M_y_^+^ nH_2_O)(Al_2−y_Mg_y_) Si_4_O_10_(OH)_2_, where the octahedral cation is predominantly Al^3+^. Montmorillonite has gained widespread interest in different fields due to its low cost, abundant resources, tunable 2D layered structure, unique ionic conductivity, and high specific surface area [10,11,12,13]. On the other hand, the saponite is a trioctahedral smectite whose predominant octahedral cation is Mg^2+^, and its general formula could be described as (Mg_3−y_(Al, Fe)_y_)(Si_4−x_Al_x_)O_10_(OH)_2_ [14,15,16]. Recent advances in modifying saponite through surface engineering and intercalation chemistry have enhanced its functionalities for various applications [17]. 

The versatility for CO_2_ adsorption of clay minerals is due to their ability to modify their structure in response to chemical and physical changes [7]. Phyllosilicates, in particular, can be chemically treated to enhance their adsorption capacity through processes such as cation exchange and surface functionalization. For example, acid activation of clays can increase their surface area and improve the accessibility to their adsorption sites [18,19]. Montmorillonite and saponite stand out not only for their adsorption properties but also for their ability to swell and shrink, allowing them to trap CO_2_ in their interlayer spaces [20]. This behavior is especially useful in carbon capture and storage technologies, where the ability to store large volumes of gas in small spaces is required. Additionally, these minerals can be easily regenerated through thermal desorption, allowing for their reuse in multiple capture cycles.

Advances in nanoscience and nanotechnology have also opened new possibilities for the use of clays in CO_2_ capture. The creation of clay nanocomposites with polymers and other materials can significantly improve their adsorption capacity and mechanical stability [21]. These hybrid materials can be designed to have specific properties, such as high selectivity for CO_2_ in the presence of other gases, making them ideal for large-scale industrial applications, for example, converting captured CO_2_ into value-added products [21].

In this work, the adsorbent properties of two of the most common clay minerals of the smectite group, montmorillonite and saponite, after being subjected to different types of treatments, are studied. For that, novel microwave acid-assisted treatments, which involve the use of lower acid concentrations and allow for reduced treatment times, are applied to clay minerals to enhance their adsorption capacity [18,19]. On the other hand, Porous Clay Heterostructures (PCHs) are prepared using these clays as starting materials. These new materials are known to have an expansion of the interlayer space up to 45.8 Å, producing notable increases in the specific surface area [22,23]. The novelty of the present work is focused on the use of two treatments (microwave-assisted acid treatment and the formation of PCHs) to generate a greater surface area and microporosity, which is directly related to the ability to capture CO_2_. In this work, the synergistic effect of both methodologies was evaluated in two types of smectites (montmorillonite and saponite).

## 2. Materials and Methods

### 2.1. Porous Clay Heterostructures Synthesis and Microwave Acid-Assisted Treatments

Two selected smectites from the Madrid basin, supplied by TOLSA, S.A (Madrid, Spain), one dioctahedral and one trioctahedral, were utilized in this study. The dioctahedral smectite corresponds to a montmorillonite sample, while the trioctahedral smectite is a saponite, henceforth referred to as Mont and Sap, respectively. All chemicals were obtained from Merck KGaA (Darmstadt, Germany) and used as received without further purification. Ion exchange column-deionized (DI) water was employed for all syntheses.

Porous Clay Heterostructures (PCH) were synthesized by slightly modifying the procedure detailed in [23]. Firstly, 2.5 g of both the preliminary montmorillonite and saponite were treated with 9 g of cetyltrimethylammonium bromide (CTMABr) in 100 mL of pure 1-propanol (anhydrous, 99.9% VWR). The solution was stirred for 3 days and later was filtered and washed in distilled water until neutral pH to promote the intercalation of CTMA^+^-cations in the interlayer spacing of both smectites. Then, the swelling smectites were re-suspended with 250 mL of H_2_O for 24 h. After this time, a solution of 0.88 g of hexadecylamine, used as a co-surfactant, in 25 mL of the 1-propanol solution was added to the mother suspension and stirred for 24 h. Preparation of the PCHs was carried out by adding a silicon source of tetraethyl orthosilicate (TEOS) diluted in 1-propanol (50 vol.%). The resulting suspension was stirred at room temperature for 3 days and dried overnight at 80 °C. Finally, in order to remove the organic matter, PCHs were calcined at 550 °C for 6 h, using a rate of 1 °C min^−1^. Hereafter, samples will be labeled as Mont-PCH and Sap-PCH (Figure 1).

Finally, 5 g of Mont, Sap, Mont-PCH, and Sap-PCH, previously heated up to 60 °C for 48 h, were treated with 50 mL of HNO_3_ 0.2 N for up to 16 min under 1000 W of microwave radiation in an open glass cylinder reactor [18]. The microwave irradiation was applied in a discontinuous form, following the methodology described by Korichi et al. (2012) [24] and Franco et al. (2016) [18], promoting more uniform heating compared to traditional methods. After each 2 min of microwave irradiation, the suspensions were cooled for 5 min to avoid the loss of the solution (Figure 1). Then, samples were washed and dried at 80 °C for 48 h. The time shown in each sample refers to the microwave-assisted treatment time exclusively.

### 2.2. Structural, Chemical, Textural and Morphological Characterization

The starting smectites and the resulting PCH materials were characterized by X-ray diffraction (XRD) using an X’Pert Pro MPD automated diffractometer (PANalytical B.V., Almelo, The Netherlands.) equipped with a Ge (111) primary monochromator (monochromatic CuK_α1_ radiation) and an X’Celerator detector. Diffractogram patterns were obtained over the 2θ range of 2–65° with a 0.0133° step size during a measurement time of 33 min per pattern.

The chemical analysis, focusing on major elements, was conducted using the MagiX X-ray fluorescence (XRF) spectrometer from PANalytical B.V. (Almelo, The Netherlands). This state-of-the-art instrument allows for precise and accurate determination of elemental compositions, ensuring reliable and reproducible results essential for the characterization of the smectite samples.

Fourier-transform infrared spectroscopy (FT-IR) spectra were collected using a Harrick HVC-DRP cell (Harrick Scientific, Pleasantville, NY, USA) fitted to a Varian 3100 FT-IR spectrophotometer (Varian, Inc., Palo Alto, CA, USA). Interferograms consisted of 200 scans. Spectra were collected using a KBr spectrum as a background. About 30 mg of finely ground clay samples were placed in the sample holder.

The nitrogen adsorption–desorption isotherms of the unmodified and treated clays were measured at −196 °C using Micromeritics ASAP 2020 equipment (Micromeritics, Norcross, GA, USA) static volumetric technique). All samples were degassed under high vacuum (∼5 × 10^−6^ bar) at 423 K for 12 h, and the measured weight loss showed full removal of the solvents. The N_2_ isotherms were analyzed by the Brunauer–Emmett-Teller (BET) method [25], whereas the surface area of the external surface and the micropores area were calculated using the t-plot method. Carbon dioxide sorption isotherms were also measured using Micromeritics ASAP 2020 equipment. Given that at high-pressure CO_2_ liquefies, it was not possible to reach the maximum absorption experimentally. Accordingly, in this study, the adsorption analysis was limited to the relative pressure at which CO_2_ liquefies (P/P_o_ = 0.016). The samples were degassed at 463 K for 2 h and the CO_2_ adsorption isotherms were obtained at 273 K and analyzed by the Dubinin–Radushkevich (DR) method [26].

The morphology of the samples was observed using a Scanning Electron Microscope (SEM) with a JEOL JSM-6490LV combined with X-ray energy dispersive spectroscopy (EDX) (JEOL Ltd., Tokyo, Japan). Samples were previously gold-sputtered (10 nm thick) in order to avoid charging the surface. 

## 3. Results and Discussion

### 3.1. Structural Characterization of Starting and Resulting Materials

Figure 1 displays the XRD patterns for the two starting samples of montmorillonite (Mont) and saponite (Sap). A detailed analysis of the XRD patterns shows that the Sap sample contains, in addition to saponite, minor amounts of quartz and dolomite, whereas the Mont sample includes montmorillonite and scarce quartz, plagioclase, and gypsum. In the Figure, the characteristic 001, 002, and 060 reflections for montmorillonite are marked, which correspond with the interplanar distances of 12.63, 6.30, and 1.50 Å, respectively, while for saponite, the 001 and 060 reflections are indicated with spacings of 12.74 and 1.52 Å, respectively. These values are consistent with those reported in the literature [27,28]. On the other hand, the lower intensity of the diffraction peaks in the Sap suggests a lower crystallinity for this sample.

After the acid treatment, no remarkable differences in the non-basal reflections of montmorillonite (Mont-H), centered at 20° and ~63°, respectively, apart from the evident disappearance of gypsum diffraction peaks, have been detected (Figure 2a). This result suggests that the montmorillonite sample has great resistance to this type of treatment, as already suggested by the persistence in the contents of Si and Al in the chemical analyses carried out throughout the treatment (see Section 3.2). The only noteworthy observed differences are the slight shifting to smaller angles and the broadening of the (001) diffraction peak, indicating that some changes in the interlayer position occur during the treatment. These changes in the interlayer positions may be due to the exchange of interlayer cations for H^+^ and the degree of hydration of the interlayer space since the acid treatment is carried out in an aqueous solution that generates homogeneous hydration of the particles. On the contrary, in the case of the Sap sample (Sap-H), this acid treatment provoked a decrease in intensity in the reflections centered between 20° and 30° as well as in the band centered at 61°, compared with the initial saponite. These bands are replaced by a wide band centered at 25° due to the appearance of an amorphous phase. Moreover, the intensity of the diffraction band centered at 7°, corresponding to the basal reflection (001), progressively decreases as the time of exposure to the acid treatment increases. After 16 min of exposure, the band has almost disappeared. These results show that, unlike montmorillonite, saponite is strongly altered by the acid treatment (Figure 2b). The different behavior is attributed to the nature of the octahedral cation. In the saponite structure, Mg^2+^ is highly soluble in acidic conditions, whereas, in montmorillonite, Al^3+^ is insoluble [18].

The influence of the microwave acid-assisted treatment on the structure of the pillared materials, both Mont-PCH and Sap-PCH, was also investigated (Figure 2c,d). So, for the Mont series, peaks corresponding to non-basal reflections (between 19°–25° and ~63°) are preserved in both as-synthesized Mont-PCH and the pillared samples subjected to the acid attack (Mont-PCH-H), despite the intensity of these bands being lower than in the case of the pillared clay. However, the band corresponding to the basal reflection (around 7°) practically disappears, suggesting that the sample has been heavily delaminated after PCH synthesis. It is noteworthy that no differences are observable between Mont-PCH and Mont-PCH-H, implying the silica pillars have not been modified during the process (Figure 2c). In the case of Sap-PCH, the (020) reflection is preserved in the as-synthesized material, although a remarkable broadening of the spectra is displayed from ~20° to ~30°. The band corresponding to the basal reflection at 7° is not present after PCH synthesis, revealing that the basal space in Sap-PCH is no longer defined. The persistence of the (060) reflection (~61°, ~1.52Å) denotes that Sap-PCH still preserves the trioctahedral nature. The patterns of the Sap-PCH-H samples resemble that of the Sap-PCH, although they also show a progressive decrease in the intensity of the (060) reflection (~61°, ~1.56Å) as the treatment time progresses. This is evident after 16 min of acid attack, which led to the formation of a highly delaminated phase, a new type of material whose structure retains the tetrahedral sheets of the starting smectites but keeps the interlaminar space of the natural smectites inside the layer between two adjacent tetrahedral sheets, leaving the previous octahedral sheets of the 2:1 layers as the new interlayer space. These suggest that this new kind of material could be named “reversed smectite”. Delamination results from the dissolution of the octahedral layer, which ultimately triggers the formation of monolayer structures (Figure 3). This feature has not been observed in the modified montmorillonite, evidencing a clearly different behavior during both the acid treatment and pillared synthesis and consequent structural modification in saponite (Figure 2).

### 3.2. Chemical Analysis

The chemical compositions of the natural smectites and the materials resulting from different treatments obtained by X-ray fluorescence are shown in Table 1. These data indicate the predominance of Al in octahedral positions in montmorillonite and Mg in saponite. Figure 4 illustrates the changes in Al_2_O_3_, MgO, and Fe_2_O_3_ content relative to the % SiO_2_ in the materials throughout the microwave-assisted acid treatments.

For saponite, acid treatment results in a gradual and significant decrease in the proportion of Mg, while the percentage of SiO_2_ increases steadily with treatment time. This indicates that the acid treatment effectively removes other components, thereby concentrating SiO_2_. For example, the SiO_2_ content rises from 49.75% in the original sample to 65.44% after 16 min of treatment. In contrast, the percentage of Al_2_O_3_ remains relatively constant with minor variations, ranging between 6.64% and 6.80%. The percentage of MgO decreases significantly with acid treatment time, indicating the solubility of MgO in the acid used dropping from 20.37% to 9.27%. The percentage of Fe_2_O_3_ shows a slight but not very significant increase with acid treatment, rising from 2.44% to 3.09%. Other components, including Na_2_O, K_2_O, and CaO, decrease considerably, particularly Na_2_O and CaO, suggesting these oxides are more soluble in the acid treatment. For instance, the Na_2_O content decreases from 2.47% to 0.39%, and the CaO content decreases from 2.15% to 0.11%. Similar behavior is observed for Sap-PCH with acid treatment time. The percentage of SiO_2_ increases significantly, rising from 76.59% to 83.84% after treatment. The percentage of MgO decreases with acid treatment time, though to a lesser extent than in saponite, reducing from 8.54% to 3.48%. The percentage of Fe_2_O_3_ decreases slightly, from 1.06% to 0.71%. Other components, such as Na_2_O, K_2_O, and CaO, also decrease significantly. For example, the Na_2_O content decreases from 0.18% to 0.11%, and the CaO content decreases from 0.77% to 0.04%.

In the case of montmorillonite, the concentrations of elements in octahedral positions, mostly Al, remain constant throughout the treatment, ranging between 18.13% and 18.84%. Similarly, Fe_2_O_3_ remains constant with minor variations, ranging between 4.18% and 4.16%. The percentage of SiO_2_ increases slightly with acid treatment, rising from 58.14% to 60.00%, while the percentage of MgO decreases slightly, from 2.91% to 2.72%. Other components, such as Na_2_O, K_2_O, and CaO, decrease significantly, especially Na_2_O, which drops from 2.99% to 0.80%, and CaO, which decreases from 1.82% to 0.55%. Similar behavior is observed for Mont-PCH. Building upon this, the SiO_2_ content increases in all samples with acid treatment, indicating that other components are removed, thereby concentrating SiO_2_. Both Al_2_O_3_ and Fe_2_O_3_ remain quite constant with only slight variations. In contrast, MgO decreases significantly in saponite, suggesting its solubility in the acid. Other oxides, such as Na_2_O, K_2_O, and CaO, decrease considerably, demonstrating their high solubility during the acid treatment.

### 3.3. FTIR Analysis

Since the infrared spectroscopic study of acid-treated smectites has been discussed in previous works [18,19], in this section, we will focus on the infrared spectroscopic study of PCHs and new materials derived from acid activation of PCHs: ‘reversed smectites’.

The evolution of the structure of these materials has also been studied by FTIR. Figure 5 shows the FTIR spectra of the initial clays (black and blue lines, respectively). As can be observed, the Mont spectra show, in the region between 4000 and 3000 cm^−1^, two bands at 3628 and 3450 cm^−1^, as well as a weak band at 3265 cm^−1^. The band at 3628 cm^−1^ is related to Al(OH)Al-stretching vibrations [30], whereas the bands at 3450 and 3265 cm^−1^ can be assigned to water-stretching vibrations in both the interlayer space and the external surface of the particles [31,32]. On the other hand, the band at 1639 cm^−1^ is classically assigned to the deformation vibrations of water molecules [31]. Bands between 1500 and 1100 cm^−1^ are typically interpreted as the Si–O stretching region of FTIR spectra. The sample studied provided an intense band at 1011 cm^−1^, with the participation of two smaller bands at 1043 and 1113 cm^−1^, the latter ascribed to the in-plane Si–O stretching modes [33]. Smaller bands are present at 914 and 882 cm^−1^ due to the deformation of the Al–OH and Mg–OH groups, respectively [31]. Regarding the Sap sample, in the OH stretching bands, two bands appear at 3618 and 3451 cm^−1^, accompanied by two weak shoulders at 3677 and 3265 cm^−1^. The band at 3677 cm^−1^ is due to the νOH of isolated Si–OH groups located on the external surface of laminar particles [34]. The band at 3618 cm^−1^ is assigned to the OH stretching mode of structural Mg(OH)_2_ groups in the octahedral sheet of the saponite structure [35,36]. Bands at 3451 and 3265 cm^−1^ result from H_2_O molecules located in the interlayer space and the external surface of saponite particles [30]. In the 1700–850 cm^−1^ region of the saponite FTIR spectrum, the band at 1639 cm^−1^ can be attributed to water bending vibrations [31], whereas the prominent peak at 1011 cm^−1^ corresponds to the Si-O stretching band. Finally, the small band located at 882 cm^−1^ points to the deformation of the Mg–OH groups [31].

Figure 5a,b include FTIR spectra for the most representative materials of the Mont and Sap series, including the initial clays for comparative purposes. Both spectra of the initial montmorillonite and after 16 min of acid treatment (Mont-H 16 min) are virtually superimposable, implying that the acid treatment does not affect the structure of montmorillonite (Figure 5b). In contrast, after 16 min of acid treatment of the initial saponite, the bands corresponding to OH groups (at 3700 and 3600 cm^−1^) were practically wiped out, and a new band at 1024 cm^−1^ appears due to the formation of amorphous silica. On the other hand, PCH synthesis conditions cause strong modifications in the FTIR spectra of montmorillonite. These modifications affect both the vibrations of the OH groups and the Si–O bonds. There is a remarkable decrease in the intensity of bands attributed to OH groups, suggesting that the insertion of the silica columns modifies the stretching modes of the octahedral sheets. Moreover, the bands corresponding to the Si–O groups in the region between 1500 and 1100 cm^−1^ are shifted towards higher wave numbers, and a shoulder appears at ~1250 cm^−1^. It is noteworthy that the acid treatment hardly modified the FTIR spectra of the pillared montmorillonite (Figure 5b). Similarly, the insertion of silica columns into the interlayer space of saponite produces modifications similar to those observed in the montmorillonite material, including the decrease and smoothing of the bands of OH groups and the displacement of the Si–O stretching bands. The loss of the band at 1400 cm^−1^ is attributed to the probable elimination of dolomite observed in the starting saponite. Contrary to montmorillonite, acid treatment of the pillared saponite produced a slight modification of the Si–O band.

### 3.4. Morphological Characteristics of Mont and Sap Series

Figure 6 shows SEM images comparing Mont, Sap, Mont-PCH, and Sap-PCH samples, both before and after microwave-assisted acid treatment. The initial Mont sample (Figure 6a) presents relatively compact and agglomerated particles with rough surfaces. After acid treatment (Figure 6e), the montmorillonite particles appear to have undergone a partial dissolution, reducing the size of individual particles. The pillared montmorillonite (Figure 6b) without acid treatment shows a less compact and agglomerated structure compared to Mont, with increased porosity due to the introduction of pillars. When the pillared montmorillonite is treated with acid (Figure 6f), there is significant disintegration of the particles, resulting in a highly porous structure with smaller, less defined particles compared to Mont-PCH.

On the other hand, the initial Sap sample presents particles with a well-defined morphology (Figure 6c), exhibiting less rough surfaces compared to Mont. After acid treatment (Figure 6g), the saponite shows a more porous structure with particles that have more irregular edges, indicating partial dissolution and restructuring of the original particles. The pillared saponite (Figure 6d), without acid treatment, displays a more open structure with more separated laminae compared to the initial saponite. The pillaring appears to increase the interlaminar distance, resulting in a more porous structure. With additional acid treatment (Figure 6h), the pillared saponite shows significantly altered morphology, with a much more porous structure and fragmented particles. The laminae appear to have separated even further, and the particles are smaller and less defined.

Thus, both montmorillonite and saponite show an increase in porosity and a reduction in particle size after acid treatments, with a more pronounced effect observed in the pillared samples. The acid-treated particles are smaller and less defined compared to their untreated counterparts, and pillaring followed by acid treatment exacerbates this effect. Additionally, SEM images reveal that the surface texture of the samples becomes significantly rougher after acid treatment. This increased roughness likely enhances adsorption properties by providing more active sites for gas interactions. Furthermore, the introduction of pillars not only increases the interlaminar distance but also appears to reinforce the structural integrity of the smectites, preventing collapse under acidic conditions. These morphological and structural changes underscore the impact of acid and pillaring treatments in improving the adsorption properties of smectites, which is crucial for applications in CO_2_ capture and other adsorption processes.

### 3.5. Textural Properties of the Materials

Table 2 compiles the specific surface areas determined from the N_2_-adsorption isotherms using the BET method [25] (Figure 7). The starting montmorillonite has a surface area (S_BET_) of 49 m^2^ g^−1^, with 24 m^2^ g^−1^ attributed to the micropore surface (S_micro_) and 25 m^2^ g^−1^ to the external surface (S_ext_). Saponite, despite being isostructural with montmorillonite, achieves an S_BET_ of 128 m^2^ g^−1^, with 76 m^2^ g^−1^ corresponding to S_micro_ and 52 m^2^ g^−1^ to S_ext_. The differences in adsorption surface values between the two smectites can be attributed to the larger particle size of montmorillonite, as suggested by the XRD data (Figure 1). The smaller particle size of saponite, combined with higher delamination, generates a “house of cards” structure, where the higher proportion of micropores is attributed to the presence of small, poorly ordered cavities in the clay structure.

The microwave-assisted acid treatment has minimal impact on the textural properties of the raw montmorillonite (S_BET_ varies from 49 to 62 m^2^ g^−1^) because the chemical composition is hardly modified after the treatment (Table 1 and Figure 4). This resistance to acid treatment is also confirmed by other techniques, such as XRD (Figure 2) and FTIR spectra (Figure 5), as both diffractograms and spectra are very similar to those obtained for the starting smectites. Conversely, acid treatment of saponite causes a drastic change in textural properties. The S_BET_ value increases from 128 to 324 m^2^ g^−1^ after only 8 min of treatment due to a simultaneous increase in both the micropore and external surface areas. These results demonstrate that the chemical composition of the octahedral layer significantly influences the efficiency of the acid treatment. The Al^3+^ species located in the octahedral layer of montmorillonite are insoluble under these acidic conditions, thereby inhibiting modifications in its texture and structure. In contrast, Mg^2+^ species are much more soluble under similar conditions, promoting higher reactivity in saponite [18].

Smectites can accommodate small molecules or bulky organic cations in their interlayer spacing, causing the smectites to swell. Previous studies have reported that porous clay heterostructures can be obtained by swelling followed by the polymerization of silicon alkoxide around bulky organic cations. This process leads to the formation of silica pillars in the interlaminar spaces after a calcination step, resulting in notable changes in the textural properties (Table 3). For instance, the S_BET_ of Mont-PCH increased dramatically compared with the unmodified sample, rising from 49 to 622 m^2^ g^−1^. This increase is primarily due to the rise in S_micro_ from 24 to 418 m^2^ g^−1^, but there is also a significant contribution from S_ext_, which varies from 25 to 204 m^2^ g^−1^. In the case of saponite, the S_BET_ shows an increase from 128 to 455 m^2^ g^−1^ between the starting and pillared clay. This increase is mainly attributed to the rise in the external surface area, S_ext_ (from 52 to 289 m^2^ g^−1^). These data suggest that Mont-PCH exhibits a higher proportion of micropores, whereas Sap-PCH shows larger pores.

Acid treatments of PCH-modified samples resulted in an increase in S_BET_ for both smectites (Table 3). For Mont-PCH, the S_BET_ value slightly increased from 622 to 660 m^2^ g^−1^ after 4 min of microwave-assisted acid treatment. This modest increase, from 418 to 445 m^2^ g^−1^, is attributed to the higher microporosity. The high stability of Mont-PCH during the microwave-assisted acid treatment is due to the strong resistance of Al^3+^ and Si^4+^ species to dissolution under these acidic conditions. Thus, the minor modification in textural properties is likely due to the digestion of some impurities and slight leaching of grain boundaries with crystalline defects, which are somewhat more susceptible to acid treatment. In contrast, the improvement in textural properties for Sap-PCH is more pronounced than that observed for Mont-PCH, with the S_BET_ reaching a maximum of 531 m^2^ g^−1^ after 12 min of acid treatment due to increases in both S_micro_ and S_ext_. The data reported in Table 1 suggests that this increase is due to the partial leaching of Mg^2+^ species located in the octahedral sheet, while the silica pillars remain relatively intact after the acid treatment. However, it is notable that prolonged treatment times slightly deteriorate the textural properties. This deterioration could be attributed to the slight collapse of the porous heterostructures.

The pore volume was also evaluated from their respective N_2_ adsorption–desorption isotherms at −196 °C (Figure 8). In the case of the pore volume, reported in Figure 8a, the microwave-assisted acid treatment has hardly any influence on porosity, confirming that this treatment does not affect the tetrahedral and octahedral layers of dioctahedral smectites. Regarding the saponite, the acid treatment causes an increase in the pore volume, obtaining a maximum value after 8 min of treatment. From this time, the pore volume progressively decreases probably due to the leaching of the Mg^2+^-species located in the octahedral sheet can cause a partial collapse of the layered structure. When the PCHs are formed, the pore volume notably increases. However, microwave-assisted acid treatment hardly influences an increase in porosity, even in the case of the Sap-PCH sample, where the acid treatment should promote partial delamination and collapse of its saponite sheets.

Similarly, the influence of microporosity after microwave-assisted acid treatment was also analyzed in Figure 8b. In this case, the trioctahedral smectite (saponite) enhances its microporosity after the microwave-assisted acid treatment. In the same way, their respective PCH (Sap-PCH) also improves its microporosity, so the partial leaching of the Mg^2+^ species in the octahedral sheet generates micropores. However, the use of prolonged acid treatment shows a clear decrease in microporosity because excessive leaching of Mg^2+^ species must cause structural collapse. In the case of the dioctahedral materials (Mont and Mont-PCH), the microporosity is very similar for all samples despite the acid treatment time.

The CO_2_ adsorption capacity was also evaluated for both series. Figure 9 shows the adsorption-desorption isotherms of CO_2_ for raw montmorillonite and saponite, their respective PCHs, and these samples subjected to microwave-assisted acid treatment. The CO_2_ adsorption capacities of the raw montmorillonite and saponite are 0.13 mmol/g (Figure 9a) and 0.45 mmol/g (Figure 9b) at 1 bar and 25 °C, respectively. This difference is primarily due to the higher microporosity of saponite, which allows it to capture a greater proportion of CO_2_ (Table 2). The acid treatment barely modified the CO_2_ adsorption capacity of the raw montmorillonite (Figure 9a, black line). These data are consistent with the high chemical stability of montmorillonite, as suggested by previous characterization data. However, the insertion of silica pillars increases the CO_2_ adsorption capacity to 0.7 mmol/g at 1 bar and 25 °C (Figure 9a, green line) due to the notable increase in microporosity, as observed in Table 3. No significant improvement was achieved after the acid treatment of Mont-PCH (Figure 9a, blue line), which is consistent with the minimal modification of textural properties, particularly microporosity, observed after microwave-assisted acid treatment.

In the case of saponite, both the acid treatment and PCH synthesis of the starting sample lead to an increase in CO_2_ adsorption up to 0.7 mmol/g at 25 °C and 1 bar (Figure 9b, green and cyan lines). However, the highest CO_2_ adsorption capacity was achieved for Sap-PCH after 16 min of microwave-assisted acid treatment, reaching up to 0.8 mmol/g at 25 °C and 1 bar (Figure 9b, green line). This increase in CO_2_ adsorption capacity can be attributed to several factors, including the poorer crystallinity of the starting Sap, the higher proportion of micropores in the formation of Sap-PCH (creating a house-of-cards structure), and the partial leaching of Mg^2+^-species located in the octahedral sheet due to the microwave-assisted acid treatment. However, the obtained results are moderate compared to fibrous phyllosilicates; for example, natural sepiolite adsorbs 1.6 mmol of CO_2_ per gram at 25 °C and 1 bar of pressure [37] due to the presence of small nanocavities with dimensions suitable for capturing CO_2_ in these nanochannels.

The CO_2_ adsorption generally takes place through physical or chemical sites. The absence of basic sites discards the presence of chemical adsorption sites, so the CO_2_ adsorption must be eminently physical. In this case, the adsorption generally occurs in the micropores due to the strong quadrupole moment of the CO_2_ molecules [37,38]. Figure 10 represents the correlation between the micropore volume and the CO_2_ adsorption capacity. The obtained results report that those samples with greater microporosity are also the materials with the greatest CO_2_ adsorption capacity. However, this correlation is not linear, probably because the micropore size influences the adsorption capacity. On the other hand, the presence of silanol groups, siloxanes, or acid sites should also influence the adsorption capacity.

The comparison of the CO_2_ adsorption capacity with the data reported in the literature shows how the values are variable for the raw clays. These data depend on the crystallinity of the starting smectite. As was indicated above, the microwave-assisted acid treatment improves the CO_2_ capture due to an increase in the microporosity. In the same way, the formation of PCH also produces an increase in the CO_2_ adsorption capacity due to a notable increase in porosity due to the formation of pillars and a partial delamination of the smectite. Thus, the formation of PCH and the use of smectite sensitive to acid treatments such as saponite results in a material with greater microporosity, which is directly related to a greater CO_2_ capture capacity, as shown in Table 4.

These findings suggest potential applications of this nanomaterial in electrochemical devices for converting captured CO_2_ into value-added products, offering a sustainable solution for CO_2_ utilization. This progress is crucial in the ongoing search for efficient and cost-effective solutions for carbon capture and storage, highlighting the potential for further enhancements in the adsorption capabilities of these materials. Such advancements not only contribute to mitigating climate change but also promote the development of innovative technologies for transforming CO_2_ into useful compounds, thereby addressing both environmental and economic challenges.

## 4. Conclusions

This study provides a comprehensive analysis of the N_2_ and CO_2_ adsorption capacities of two types of smectites—dioctahedral montmorillonite and trioctahedral saponite—before and after acid treatment and silica pillar insertion (PCHs). Our results demonstrate that microwave-assisted acid treatment causes significant structural changes in saponite within minutes, including the dissolution of the octahedral layer and the formation of amorphous silica precipitates. These changes enhance the material’s surface area and porosity, which are crucial for CO_2_ adsorption. In contrast, montmorillonite remains structurally stable under the same conditions, indicating a strong resistance to acid attack. Additionally, the insertion of silica pillars successfully increased the CO_2_ adsorption capacity in both smectites. For montmorillonite, this increase is predominantly due to a rise in micropore volume, while for saponite, the formation of Sap-PCH monolayers or “reversed smectites” leads to a significant surface area expansion. Notably, the study reveals, for the first time, the improved adsorption capacity of materials formed after acid digestion of pillared clays (PCHs), particularly saponite. This finding not only highlights the effectiveness of our novel microwave-assisted acid treatment but also opens up new possibilities for using these materials in sustainable CO_2_ capture and conversion technologies. These findings, drawn directly from our experimental results, underscore the potential of acid-treated and pillared smectites in advanced environmental applications, setting the foundation for future explorations in electrochemical devices for CO_2_ management.

## Data Availability

The original contributions presented in the study are included in the article, further inquiries can be directed to the corresponding authors.

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
