# Peer review of "Reversed Mg-Based Smectites: A New Approach for CO2 Adsorption"

_nanomaterials, 2024, doi:10.3390/nano14181532_

Round 1

Reviewer 1 Report

Comments and Suggestions for Authors

The article “Reversed Mg-based smectites: a new approach for CO2 adsorption” by Francisco Franco et al is devoted to the study of carbon dioxide CO2 sorption by clay materials (montmorillonite and saponite). As is known, montmorillonite and saponite are capable of exhibiting high sorption properties due to their porous structure. The work shows that acid modification of saponite according with ultrasound treatment leads to an increase of its adsorption capacity with respect to CO2 molecules.

The article is an interesting and logical study and can be published in the Nanomaterials journal after revision.

- Please, emphasize the novelty of work. From the last paragraph in the Introduction, it is difficult to understand how the present study differs from others.

- What do the authors mean by the words “novel microwave acid assisted treatments”? (line 74)

- On the line 101 there is an unnecessary repetition of the names of the Mont and Sap samples.

- It is worth paying attention to the correct spelling of subscripts in chemical formulas.

- What do the lines 60 di and 60 tri mean in the diffraction pattern (Fig. 1)? The authors need to better identify the phases corresponding to montmorillonite and saponite. The diffraction pattern contains a large number of impurity phases, so it is difficult to understand. It will also be useful for the reader to determine the interplanar distance, since this is a really important parameter for clarifying the sorption mechanisms.

- Figure 5a with the original clay spectra is redundant, since their spectra are shown in Fig. 5a and 5b. Work 10.1134/S0036023623600296 will be useful to the authors for comparing the IR spectra and X-ray diffraction data with the literature.

- The provided list of references contains a fairly large number of outdated references.

Author Response

Comments 1: Please, emphasize the novelty of work. From the last paragraph in the Introduction, it is difficult to understand how the present study differs from others.

Response 1: Thank you for your insightful feedback. To address your concern, we have revised the last paragraph of the Introduction to better emphasize the novelty of our work. The updated section now clearly outlines how our study differs from previous research. Specifically, we have added the following text:

The novelty of the present work is focused on the use of two treatments (microwave-assisted acid treatment and the formation of PCHs) to generate a greater surface area and microporosity, which is directly related to the ability to capture CO2. In this work, the synergistic effect of both methodologies was evaluated in two types of smectites (montmorillonite and saponite).

Comments 2: What do the authors mean by the words “novel microwave acid assisted treatments”? (line 74)

Response 2: Thank you for your insightful question. By "novel microwave acid assisted treatments," we are referring to treatments where microwave energy is used in conjunction with a reduced concentration of acid. Additionally, these treatments involve shorter processing times compared to traditional methods. The term "novel" highlights the innovative approach of combining microwaves with these optimized conditions to enhance efficiency and effectiveness.

We have included the following sentence: “novel microwave acid-assisted treatments, which involve the use of lower acid concentrations and allow for reduced treatment times” to clarify the term.

Comments 3: On the line 101 there is an unnecessary repetition of the names of the Mont and Sap samples.

Response 3: Thank you for pointing that out. We appreciate your careful review. We have removed the unnecessary repetition of the Mont and Sap sample names from the text.

Comments 4: It is worth paying attention to the correct spelling of subscripts in chemical formulas.

Response 4: Thank you for your attention to detail. We have carefully reviewed the text and ensured that the subscripts in all chemical formulas are correctly formatted.

Comments 5: What do the lines 60 di and 60 tri mean in the diffraction pattern (Fig. 1)? The authors need to better identify the phases corresponding to montmorillonite and saponite. The diffraction pattern contains a large number of impurity phases, so it is difficult to understand. It will also be useful for the reader to determine the interplanar distance, since this is a really important parameter for clarifying the sorption mechanisms.

Response 5: Thank you for your valuable feedback. The lines labelled 060 di and 060 tri in the diffraction pattern refer to the 060 reflection of dioctahedral and trioctahedral smectites, respectively. As clay minerals often exhibit low crystallinity and tend to contain impurities, these impurities are typically more crystalline, which complicates the identification of the clay minerals solely by diffraction. We have included the following sentence:

In the figure, the characteristic 001, 002 and 060 reflections for montmorillonite are marked which correspond with the interplanar distances of 12.63, 6.30, and 1.50 Å, respectively, while for saponite, the 001 and 060 reflections are indicated with spacings of 12.74 and 1.52 Å, respectively. These values are consistent with those reported in the literature [27,28].

And the following references:

[27] Carretero, M. I.; Pozo, M.; Sánchez, C.; García, F. J.; Medina, J. A.; Bernabé, J. M. Comparison of Saponite and Montmorillonite Behaviour During Static and Stirring Maturation with Seawater for Pelotherapy. Appl. Clay Sci. 2007, 36(1–3), 161–173. https://doi.org/10.1016/j.clay.2006.05.010

[28] Christidis, G.E.; Koutsopoulou, E. A Simple Approach to the Identification of Trioctahedral Smectites by X-ray Diffraction. Clay Miner. 2013, 48(5), 687–696

Comments 6: Figure 5a with the original clay spectra is redundant, since their spectra are shown in Fig. 5a and 5b. Work 10.1134/S0036023623600296 will be useful to the authors for comparing the IR spectra and X-ray diffraction data with the literature.

Response 6: Thank you for your helpful comment. We appreciate your suggestion and have modified Figure 5 to avoid redundancies. Additionally, have adjusted the colours in Figure 5 to make the lines for each sample, including the Sap-PCH and SAP-PCH-H samples, more easily distinguishable 5 following the advice of Referee 1

Comments 7: The provided list of references contains a fairly large number of outdated references.

Response 7: Thank you for your feedback. We acknowledge the importance of including up-to-date references. While some of the references in our list are foundational and provide essential background to the current study, we understand the need to balance these with more recent literature. We have reviewed and updated the reference list to include more recent studies, ensuring that our work is grounded in both historical and current research developments.

Reviewer 2 Report

Comments and Suggestions for Authors

In this paper, the CO2 adsorption capacity of various clay minerals and their treated products was investigated, especially the CO2 adsorption capacity of montmorillonite and saponite. The reaction time of acid treatment was significantly reduced by microwave-assisted acid treatment. Moreover, silica pillars were inserted into montmorillonite to form a material with a large specific surface area, which improved the adsorption capacity of the material for CO2. I agree that this article should be published on nanomaterials, but there are still some problems in the article, so I suggest that it be published after the problems are solved.

1. There are some formatting problems in the paper, such as two periods in the last sentence of the abstract, and the format of the subscript of some chemical formulas is not accurate. I hope the author can check it carefully.

2. The lines of Sap-PCH and SAP-PCH-H samples in Figure 5c are not clear, so I hope the author can separate them and mark them more clearly.

3. As for the description of Figure 6e, the author said that after acid treatment, it shows a porous structure, but the aperture of Figure 6e is not obvious, I hope the author can check it carefully.

4. On the premise of conclusion, the author mentioned that this material can convert CO2 into value-added products in electrochemical devices. The material prepared by the author has a high CO2 adsorption capacity, but too strong CO2 adsorption capacity may be unfavorable to the chemical transformation of CO2. It is hoped that the author can modify this sentence.

5. In the conclusion, the author mentioned that this study can provide a solution for carbon conversion, but the whole article talked about the adsorption of CO2, and did not make a detailed discussion of carbon conversion. I hope the author considers this sentence carefully.

6. The author has made experiments on the ability of the material to CO2 adsorbed, but has not characterized the ability of the material to CO2 desorbed. It is hoped that relevant experiments can be added to illustrate the CO2 desorption ability of the material.

Comments on the Quality of English Language

In this paper, the CO2 adsorption capacity of various clay minerals and their treated products was investigated, especially the CO2 adsorption capacity of montmorillonite and saponite. The reaction time of acid treatment was significantly reduced by microwave-assisted acid treatment. Moreover, silica pillars were inserted into montmorillonite to form a material with a large specific surface area, which improved the adsorption capacity of the material for CO2. I agree that this article should be published on nanomaterials, but there are still some problems in the article, so I suggest that it be published after the problems are solved.

1. There are some formatting problems in the paper, such as two periods in the last sentence of the abstract, and the format of the subscript of some chemical formulas is not accurate. I hope the author can check it carefully.

2. The lines of Sap-PCH and SAP-PCH-H samples in Figure 5c are not clear, so I hope the author can separate them and mark them more clearly.

3. As for the description of Figure 6e, the author said that after acid treatment, it shows a porous structure, but the aperture of Figure 6e is not obvious, I hope the author can check it carefully.

4. On the premise of conclusion, the author mentioned that this material can convert CO2 into value-added products in electrochemical devices. The material prepared by the author has a high CO2 adsorption capacity, but too strong CO2 adsorption capacity may be unfavorable to the chemical transformation of CO2. It is hoped that the author can modify this sentence.

5. In the conclusion, the author mentioned that this study can provide a solution for carbon conversion, but the whole article talked about the adsorption of CO2, and did not make a detailed discussion of carbon conversion. I hope the author considers this sentence carefully.

6. The author has made experiments on the ability of the material to CO2 adsorbed, but has not characterized the ability of the material to CO2 desorbed. It is hoped that relevant experiments can be added to illustrate the CO2 desorption ability of the material.

Author Response

Comments 1: There are some formatting problems in the paper, such as two periods in the last sentence of the abstract, and the format of the subscript of some chemical formulas is not accurate. I hope the author can check it carefully.

Response 1: Thank you for your attention to detail. We have carefully reviewed the text and ensured that the subscripts in all chemical formulas are correctly formatted.

Comments 2: The lines of Sap-PCH and SAP-PCH-H samples in Figure 5c are not clear, so I hope the author can separate them and mark them more clearly.

Response 2: Thank you for your helpful comment. We appreciate your suggestion and have adjusted the colours in Figure 5 to make the lines for each sample, including the Sap-PCH and SAP-PCH-H samples, more easily distinguishable. Additionally, we have modified Figure 5 according to the feedback from Referee 1, including the removal of Figure 5a to avoid redundancies.

Comments 3: As for the description of Figure 6e, the author said that after acid treatment, it shows a porous structure, but the aperture of Figure 6e is not obvious, I hope the author can check it carefully.

Response 3: Thank you for your careful observation. You are correct that the porosity is not clearly visible in Figure 6e. We have revised the text to reflect this, and it now reads: "After acid treatment (Figure 6e), the montmorillonite particles appear to have undergone a partial dissolution, reducing the size of individual particles."

Comments 4: On the premise of conclusion, the author mentioned that this material can convert CO2 into value-added products in electrochemical devices. The material prepared by the author has a high CO2 adsorption capacity, but too strong CO2 adsorption capacity may be unfavourable to the chemical transformation of CO2. It is hoped that the author can modify this sentence.

Response 4: Thank you for your valuable feedback. To demonstrate that the CO2 adsorption process is reversible, we have included the CO2 desorption isotherm in Figure 9. This addition helps to illustrate that the material can effectively release adsorbed CO2, which supports its potential for use in electrochemical devices for CO2 conversion despite its strong adsorption capacity.

Comments 5: In the conclusion, the author mentioned that this study can provide a solution for carbon conversion, but the whole article talked about the adsorption of CO2, and did not make a detailed discussion of carbon conversion. I hope the author considers this sentence carefully.

Response 5: We have rephrased the conclusion to clarify that the study primarily addresses CO2 adsorption as a foundational step with the potential for future research to explore its application in CO2 conversion processes and now the conclusion says the following:

These findings, drawn directly from our experimental results, underscore the potential of acid-treated and pillared smectites in advanced environmental applications, setting the foundation for future explorations in electrochemical devices for CO2 management.

Comments 6: he author has made experiments on the ability of the material to CO2 adsorbed, but has not characterized the ability of the material to CO2 desorbed. It is hoped that relevant experiments can be added to illustrate the CO2 desorption ability of the material.

Response 6: Thank you for your valuable suggestion. We have taken your feedback into account and have modified Figure 9 to now include the CO2 desorption data. This addition helps to better illustrate the material's ability to release adsorbed CO2.

Reviewer 3 Report

Comments and Suggestions for Authors

This study examines the CO2 adsorption capacity of various clay minerals and their treated products, focusing on montmorillonite and saponite smectites. Utilizing a novel microwave-assisted acid treatment, significant reductions in reaction times are achieved. While montmorillonite shows minimal structural changes after treatment, saponite experiences substantial alterations. Additionally, the insertion of silica pillars into smectites increased CO2 adsorption capacity by creating a new type of nanomaterial with higher surface area and micropores, denoted as reversed smectite, which exhibits the highest CO2 adsorption capacity, with CO2 molecules preferentially retained inits structural surfaces. These findings suggest potential applications of this nanomaterial in electrochemical devices for converting captured CO2 into value-added products, offering a sustainable solution for CO2 utilization..

From my perspective, it is an interesting work, but it should be well revised before it can be considered.

1) Abstract.   This summary is more like a conclusion, without seeing the research necessity, or innovation, and academic value of this research. I suggest the authors condense it again, and the conclusion should not be so long.

2) line 20. “lution for CO2 utilization..” why there are two dots..?

3) 1. Introduction

Line 29,. “Atmospheric CO2 concentrations have risen from 300 ppm before the industrial revolution to over 400 ppm today.” I suggest adding a reference here. For example “Wu C, Huang Q, Xu Z, Sipra AT, Gao N, Vandenberghe LPDS, et al. A comprehensive review of carbon capture science and technologies. Carbon Capture Sci Technol 2024;11:100178.”

4). Line 34. “The capture process involves selectively storing carbon dioxide using a dry absorbent and separating it based on interactions with solid materials.” I do not think so.

Instead of fixing/storing the CO2 by adsorbent, CO2 capture is to enrich the CO2 stream for subsequent geological storage or chemical utilization.

For example, the calcium looping process. “Tian S, Yan F, Zhang Z, Jiang J. Calcium-looping reforming of methane realizes in situ CO2 utilization with improved energy efficiency. Sci Adv 2019;5:eaav5077.”

Magnesium looping process “Bork AH, Ackerl N, Reuteler J, Jog S, Gut D, Zboray R, et al. Model structures of molten salt-promoted MgO to probe the mechanism of MgCO3 formation during CO2 capture at a solid–liquid interface. J Mater Chem A 2022;10:16803–12.”

Lithium silicate looping ” Yuan W, Deng T, Chen S, He Y, Qin C. Understanding the competition between carbonation and sulfation of Li4SiO4-based sorbents for high-temperature CO2 capture. Carbon Capture Sci Technol 2022;3:100046.”

DAC cycle “Shi X, Xiao H, Kanamori K, Yonezu A, Lackner KS, Chen X. Moisture-Driven CO2 Sorbents. Joule 2020;4:1823–37.”

5) Line 42. “These phyllosilicates present very interesting “ Too colloquial.

6) The last two paragraphs of the introduction should be the most important part. The author should extract the existing research status, existing technical problems, the innovation of this work, and what kind of results are expected to be achieved.

7) 2.2. Structural, chemical, textural and morphological characterization.

Line 115. “Diffractogram patterns were obtained over the 2θ range of 2–65°with a 0.017° step size,” I do not think so, It should be 0.013303°,rather than 0.017.

8) Figure 1. and Figure 2. Since it is a dimensionless number(a.u,.), the values on the y-axis can be removed.

9) The exploration of the mechanism is too thin. It is suggested to analyze it in more detail and answer a question: why does it have such a good effect?

10) The conclusion is too long. It can be simplified. There is no need to divide it into two parts.

11) References.

The literature is too old, it is recommended to cite the latest relevant literature.

Comments on the Quality of English Language

 Minor editing of English language required.

Author Response

Comments 1: Abstract.This summary is more like a conclusion, without seeing the research necessity, or innovation, and academic value of this research. I suggest the authors condense it again, and the conclusion should not be so long.

Response 1: Thank you for your suggestions. We have revised the text accordingly:

Revised Abstract:

Addressing climate change requires transitioning to cleaner energy sources and adopting advanced CO2 capture techniques. Clay minerals are effective in CO2 adsorption due to their regenerative properties. Recent advancements in nanotechnology further improve their efficiency and potential for use in carbon capture and storage. This study examines the CO2 adsorption properties of montmorillonite and saponite, which are subjected to a novel microwave-assisted acid treatment to enhance their adsorption capacity. While montmorillonite shows minimal changes, saponite undergoes significant alterations. Furthermore, the addition of silica pillars to smectites results in a new nanomaterial with higher surface area (653 m2 g-1), denoted as reversed smectite, with enhanced CO2 adsorption capabilities, potentially useful for electrochemical devices for converting captured CO2 into value-added products.

Comments 2: line 20. “lution for CO2 utilization..” why there are two dots..?

Response 2: Thank you for your attention to detail. We have correctly formatted

Comments 3: 1. Introduction. Line 29,. “Atmospheric CO2 concentrations have risen from 300 ppm before the industrial revolution to over 400 ppm today.” I suggest adding a reference here. For example “Wu C, Huang Q, Xu Z, Sipra AT, Gao N, Vandenberghe LPDS, et al. A comprehensive review of carbon capture science and technologies. Carbon Capture Sci Technol 2024;11:100178.”

Response 3: Included

Comments 4: Line 34. “The capture process involves selectively storing carbon dioxide using a dry absorbent and separating it based on interactions with solid materials.” I do not think so.

Instead of fixing/storing the CO2 by adsorbent, CO2 capture is to enrich the CO2 stream for subsequent geological storage or chemical utilization.

For example, the calcium looping process. “Tian S, Yan F, Zhang Z, Jiang J. Calcium-looping reforming of methane realizes in situ CO2 utilization with improved energy efficiency. Sci Adv 2019;5:eaav5077.”

Magnesium looping process “Bork AH, Ackerl N, Reuteler J, Jog S, Gut D, Zboray R, et al. Model structures of molten salt-promoted MgO to probe the mechanism of MgCO3 formation during CO2 capture at a solid–liquid interface. J Mater Chem A 2022;10:16803–12.”

Lithium silicate looping ” Yuan W, Deng T, Chen S, He Y, Qin C. Understanding the competition between carbonation and sulfation of Li4SiO4-based sorbents for high-temperature CO2 capture. Carbon Capture Sci Technol 2022;3:100046.”

DAC cycle “Shi X, Xiao H, Kanamori K, Yonezu A, Lackner KS, Chen X. Moisture-Driven CO2 Sorbents. Joule 2020;4:1823–37.”

Response 4: Thank you for your valuable feedback. We have revised the text to reflect that the capture process focuses on concentrating the CO2 stream for subsequent geological storage or chemical utilization. Additionally, we have updated the manuscript to include the relevant references as suggested. The text we have included is:

The capture process aims to concentrate the CO2 stream for future geological storage or chemical use, rather than merely storing the CO2 with an absorbent

Comments 5: Line 42. “These phyllosilicates present very interesting “ Too colloquial.

Response 5: We have replaced the text for “These phyllosilicates exhibit highly intriguing physicochemical properties

Comments 6: The last two paragraphs of the introduction should be the most important part. The author should extract the existing research status, existing technical problems, the innovation of this work, and what kind of results are expected to be achieved.

Response 6: Following the advice of the reviewer, the authors have included a last paragraph in the end of the introduction section. “The novelty of the present work is focused on the use of two treatments (microwave-assisted acid treatment and the formation of PCHs) to generate a greater surface area and microporosity, which is directly related to the ability to capture CO2. In this work, the synergistic effect of both methodologies was evaluated in two types of smectites (montmorillonite and saponite).”

Comments 7: 2.2. Structural, chemical, textural and morphological characterization.

Line 115. “Diffractogram patterns were obtained over the 2θ range of 2–65°with a 0.017° step size,” I do not think so, It should be 0.013303°,rather than 0.017.

Response 7: Thank you for pointing out this error. You are correct; the correct step size should be 0.013303° instead of 0.017°. We have made this correction in the manuscript, and the updated value is now reflected in line 115. We appreciate your careful review and attention to detail.

Comments 8: Figure 1. and Figure 2. Since it is a dimensionless number(a.u,.), the values on the y-axis can be removed.

Response 8: Done

Comments 9: The exploration of the mechanism is too thin. It is suggested to analyze it in more detail and answer a question: why does it have such a good effect?

Response 9: Thank you for your suggestion. We have included a detailed representation of the microwave-assisted acid treatment mechanism in Scheme 1 of the revised manuscript to enhance clarity and facilitate understanding of this process.

Comments 10: The conclusion is too long. It can be simplified. There is no need to divide it into two parts.

Response 10: Thank you for your feedback. In response to your suggestion, we have simplified the conclusion section by condensing the content and removing the division into two parts. The revised conclusion is now more concise while still capturing the key findings and significance of our study.

This study provides a comprehensive analysis of the N2 and CO2 adsorption capacities of two types of smectites—dioctahedral montmorillonite and trioctahedral saponite—before and after acid treatment and silica pillar insertion (PCHs). Our results demonstrate that microwave-assisted acid treatment causes significant structural changes in saponite within minutes, including the dissolution of the octahedral layer and the formation of amorphous silica precipitates. These changes enhance the material's surface area and porosity, which are crucial for CO2 adsorption. In contrast, montmorillonite remains structurally stable under the same conditions, indicating a strong resistance to acid attack. Additionally, the insertion of silica pillars successfully increased the CO2 adsorption capacity in both smectites. For montmorillonite, this increase is predominantly due to a rise in micropore volume, while for saponite, the formation of Sap-PCH monolayers or "reversed smectites" leads to a significant surface area expansion. Notably, the study reveals, for the first time, the improved adsorption capacity of materials formed after acid digestion of pillared clays (PCHs), particularly saponite. This finding not only highlights the effectiveness of our novel microwave-assisted acid treatment but also opens up new possibilities for using these materials in sustainable CO2 capture and conversion technologies. These findings, drawn directly from our experimental results, underscore the potential of acid-treated and pillared smectites in advanced environmental applications, setting the foundation for future explorations in electrochemical devices for CO2 management

Comments 11: References.

The literature is too old, it is recommended to cite the latest relevant literature.

Response 11: Thank you for your feedback. We acknowledge the importance of including up-to-date references. While some of the references in our list are foundational and provide essential background to the current study, we understand the need to balance these with more recent literature. We have reviewed and updated the reference list to include more recent studies, ensuring that our work is grounded in both historical and current research developments.

Reviewer 4 Report

Comments and Suggestions for Authors

This study investigates the CO2 adsorption capacity of various clay minerals and their treated products, focusing on montmorillonite and saponite smectites. Utilizing a novel microwave-assisted acid treatment, significant reductions in reaction times are achieved. While montmorillonite shows minimal structural changes after treatment, saponite experiences substantial alterations. The insertion of silica pillars into smectites increased CO2 adsorption capacity by creating a new type of nanomaterial with higher surface area and micropores.  The authors suggest potential applications of this nanomaterial in electrochemical devices for converting captured CO2 into value-added products, offering a sustainable solution for CO2 utilization.

The paper is well written, concise; the main objective is well defined. However, there are some issues that need to be addressed before it can be considered for publication, and I recommend major revision.

1. To make it easier to follow, I suggest authors to add the flowchart of Porous Clay Heteroestructures synthesis.

2. In order to outline the scope and focus of the research and summarize the most important findings of the study, authors should re-write the Abstract, add some quantitative informations in the Abstract.

3. The novelty of this research should be clearly pointed out at the end of Introduction.

4. Please, support the XRD results with more citations, one reference is insufficient for this part of the study.

5. Please, define LOI at the spot (Table 1). Also, add a brief discussion of the obtained LOI values.

6. The values of SBET, Smicro, and Sex (Table 2) are the same as those presented in Table 2 from Ref. 16. Please indicate that the data were taken from Ref.16 if the same samples were used in these two studies.

7. The comparison with similar materials and the advantages of adsorbent used in this research should be presented.

8. On the basis of the obtained results, which of the synthesized materials shows the best performances and would be the most suitable for practical application?

9. Conclusions should be re-written. It is too general. Explain the conclusion in more detail based on the obtained results.

Author Response

Comments 1: To make it easier to follow, I suggest authors to add the flowchart of Porous Clay Heteroestructures synthesis.

Response 1: Thank you for your suggestion. We have included a flowchart of the Porous Clay Heterostructures synthesis in the revised manuscript to enhance clarity and facilitate understanding of the synthesis process.

Comments 2: In order to outline the scope and focus of the research and summarize the most important findings of the study, authors should re-write the Abstract, add some quantitative information in the Abstract.

Response 2: Thank you for your suggestions. We have revised the text accordingly to better emphasize the research necessity, innovation, and academic significance of our study:

Revised Abstract:

Addressing climate change requires transitioning to cleaner energy sources and adopting advanced CO2 capture techniques. Clay minerals are effective in CO2 adsorption due to their regenerative properties. Recent advancements in nanotechnology further improve their efficiency and potential for use in carbon capture and storage. This study examines the CO2 adsorption properties of montmorillonite and saponite, which are subjected to a novel microwave-assisted acid treatment to enhance their adsorption capacity. While montmorillonite shows minimal changes, saponite undergoes significant alterations. Furthermore, the addition of silica pillars to smectites results in a new nanomaterial with higher surface area (653 m2 g-1), denoted as reversed smectite, with enhanced CO2 adsorption capabilities, potentially useful for electrochemical devices for converting captured CO2 into value-added products.

Comments 3: The novelty of this research should be clearly pointed out at the end of Introduction.

Response 3: Following the advice of the reviewer, the authors have included a last paragraph in the end of the introduction section. “The novelty of the present work is focused on the use of two treatments (microwave-assisted acid treatment and the formation of PCHs) to generate a greater surface area and microporosity, which is directly related to the ability to capture CO2. In this work, the synergistic effect of both methodologies was evaluated in two types of smectites (montmorillonite and saponite).

Comments 4: Please, support the XRD results with more citations, one reference is insufficient for this part of the study.

Response 4: We have included these:

  1. Carretero, M. I.; Pozo, M.; Sánchez, C.; García, F. J.; Medina, J. A.; Bernabé, J. M. Comparison of Saponite and Montmorillo-nite Behaviour During Static and Stirring Maturation with Seawater for Pelotherapy. Appl. Clay Sci. 2007, 36(1–3), 161–173.
  2. Christidis, G.E.; Koutsopoulou, E. A Simple Approach to the Identification of Trioctahedral Smectites by X-ray Diffraction. Clay Miner. 2013, 48(5), 687–696

Comments 5: Please, define LOI at the spot (Table 1). Also, add a brief discussion of the obtained LOI values.

Response 5: Thank you for your feedback. We have reviewed the LOI (Loss on Ignition) data and determined that it does not provide significant additional information relevant to the discussion. Consequently, we have removed the LOI values from Table 1 and revised the manuscript accordingly.

Comments 6: The values of SBET, Smicro, and Sex (Table 2) are the same as those presented in Table 2 from Ref. 16. Please indicate that the data were taken from Ref.16 if the same samples were used in these two studies.

Response 6: Thank you for your observation. The values of SBET, Smicro, and Sext presented in Table 2 are indeed from the same samples as those in Ref. 16. This current work is an extension of that previous study conducted by our research group. We have now clearly indicated in the table captions that the data were taken from Ref. 16, as these values are foundational to the further analysis and developments presented in this manuscript.

Comments 7: The comparison with similar materials and the advantages of adsorbent used in this research should be presented.

Response 7: We have included a new table, table 4 at the end of the article:

Table 4. Comparison of CO2 adsorption capacity of several related materials

Smectite

CO2 adsorption

(mmol/g)

Conditions

Ref.

Raw smectites

Bentonite

0.14

25 °C, 1 bar

[39]

Bentonite

0.11

25 °C, 1 bar

[40]

Bentonite

0.32

45 °C, 1 bar

[20]

Montmorillonite

0.23

45 °C, 1 bar

[20]

Montmorillonite

0.16

25 °C, 1 bar

[41]

Montmorillonite

0.50

10 °C, 1 bar

[42]

Montmorillonite

0.13

25 °C, 1 bar

This work

Saponite

0.34

45 °C, 1 bar

[20]

Saponite

0.45

25 °C, 1 bar

This work

Smectites modified with acid treatment

Bentonite

0.20

25 °C, 1 bar

[18]

Bentonite

0.54

25 °C, 1 bar

[43]

Montmorillonite

0.14

25 °C, 1 bar

This work

Saponite

0.71

25 °C, 1 bar

This work

Porous clay heterostructures

Bentonite-PCH

0.64

25 °C, 1 bar

[40]

Montmorillonite-PCH

0.69

25 °C, 1 bar

This work

Saponite-PCH

0.71

25 °C, 1 bar

This work

Porous clay heterostructures modified with acid treatment

Montmorillonite-PCH

0.70

25 °C, 1 bar

This work

Saponite-PCH

0.81

25 °C, 1 bar

This work

Comments 8: On the basis of the obtained results, which of the synthesized materials shows the best performances and would be the most suitable for practical application?

Response 8: Considering the obtained results, the highest CO2 adsorption capacity is achieved when a porous clay heterostructure is formed followed by a microwave-assisted acid treatment. Both methods contribute to an increase in surface area and microporosity. Specifically, the synthesis of a porous clay heterostructure creates a "house of cards" structure with higher microporosity compared to the original smectite. Additionally, trioctahedral clays are susceptible to partial leaching of their octahedral sheets, which further enhances surface area and microporosity. Based on these findings, the sample with the most promising potential for large-scale application is Sap-PCH-H-12min.

Comments 9: Conclusions should be re-written. It is too general. Explain the conclusion in more detail based on the obtained results.

Response 9: Thank you for your constructive feedback. We have carefully considered your suggestion and have rewritten the conclusion section to provide a more detailed and specific explanation based on the obtained results. The revised conclusions now highlight the key findings from our study, emphasizing the structural changes observed in the smectites after microwave-assisted acid treatment and silica pillar insertion, as well as their impact on CO2 adsorption capacity

This study provides a comprehensive analysis of the N2 and CO2 adsorption capacities of two types of smectites—dioctahedral montmorillonite and trioctahedral saponite—before and after acid treatment and silica pillar insertion (PCHs). Our results demonstrate that microwave-assisted acid treatment causes significant structural changes in saponite within minutes, including the dissolution of the octahedral layer and the formation of amorphous silica precipitates. These changes enhance the material's surface area and porosity, which are crucial for CO2 adsorption. In contrast, montmorillonite remains structurally stable under the same conditions, indicating a strong resistance to acid attack. Additionally, the insertion of silica pillars successfully increased the CO2 adsorption capacity in both smectites. For montmorillonite, this increase is predominantly due to a rise in micropore volume, while for saponite, the formation of Sap-PCH monolayers or "reversed smectites" leads to a significant surface area expansion. Notably, the study reveals, for the first time, the improved adsorption capacity of materials formed after acid digestion of pillared clays (PCHs), particularly saponite. This finding not only highlights the effectiveness of our novel microwave-assisted acid treatment but also opens up new possibilities for using these materials in sustainable CO2 capture and conversion technologies. These findings, drawn directly from our experimental results, underscore the potential of acid-treated and pillared smectites in advanced environmental applications, setting the foundation for future explorations in electrochemical devices for CO2 management

Round 2

Reviewer 1 Report

Comments and Suggestions for Authors

The authors have improved the manuscript in accordance with the reviewer's previous comments. The study can be recommended for  publication in the Nanomaterials journal.

Author Response

We sincerely appreciate the time and effort you have invested in reviewing our manuscript. Your valuable feedback has greatly contributed to improving the quality of the article.

Reviewer 2 Report

Comments and Suggestions for Authors

the authors have revised the manuscript following all previous suggestions, I agree for its publication in the present form.

Comments on the Quality of English Language

the authors have revised the manuscript following all previous suggestions, I agree for its publication in the present form.

Author Response

(The authors gave the same response as above.)

Reviewer 3 Report

Comments and Suggestions for Authors

After revision, the manuscript has been improved significantly.

It now can be considered.

But please double-check the manuscript carefully before it been published.

1)     Figure1 and 2. Why is the font of the vertical coordinate not centered?

2)     Figure 2. The number of vertical coordinates has not been deleted. Since it is dimensionless, how can there be numbers?

3)     Table 1. Can you make it a three line Table?

Author Response

Comments 1: 1) Figure1 and 2. Why is the font of the vertical coordinate not centered?

Response 1: Thank you for your feedback. Regarding Figures 1 and 2, we appreciate your observation. We will review the alignment of the vertical axis labels and ensure that the font is properly centered for better clarity and presentation.

Comments 2: 2) Figure 2. The number of vertical coordinates has not been deleted. Since it is dimensionless, how can there be numbers?

Response 2: Thank you for pointing this out. We apologize for the confusion. In the previous submission, we mistakenly forgot to update Figure 2 as you had requested. The vertical axis should indeed be dimensionless, and we will remove the numbers in the corrected version of the figure. We appreciate your careful review and will ensure that the figure is updated accordingly.

Comments 2: 3) Table 1. Can you make it a three line Table?

Response 3: Thank you for your suggestion. We had modified Table 1 to follow the three-line table format as requested. This adjustment will be included in the revised version of the manuscript.

Please note that all revisions in the manuscript have been marked in green for easier identification.

We sincerely appreciate the time and effort you have invested in reviewing our manuscript. Your valuable feedback has greatly contributed to improving the quality of the article.

Reviewer 4 Report

Comments and Suggestions for Authors

Dear Colleagues,

I have assessed the revised manuscript ID: nanomaterials-3161379, Reversed Mg-based smectites: a new approach for CO2 adsorption”, authors Francisco Franco, Juan Antonio Cecilia, Laura Pardo, Salima Essih, Manuel Pozo, Lucía Dos Santos-Gómez, Rosario M. P. Colodrero.

The corrections made by authors significantly improved the manuscript and I find it suitable for publication in Nanomaterials.

Author Response

(The authors gave the same response as above.)
